# Comparative Transcriptome Profiling of mRNA and lncRNA of Mouse Spleens Inoculated with the Group ACYW135 Meningococcal Polysaccharide Vaccine

**DOI:** 10.3390/vaccines11081295

**Published:** 2023-07-28

**Authors:** Nan Zhu, Liping Hu, Wenlong Hu, Qiang Li, Haiguang Mao, Mengting Wang, Zhijian Ke, Lili Qi, Jinbo Wang

**Affiliations:** 1School of Biological and Chemical Engineering, NingboTech University, Qianhunan Road 1, Ningbo 315100, China; nan.zhu@aimbio.com (N.Z.); hu.liping@aimbio.com (L.H.); wenlong.hu@aimbio.com (W.H.); qiang.li@aimbio.com (Q.L.); wmt@nbt.edu.cn (M.W.); kezj@nbt.edu.cn (Z.K.); qll@nbt.edu.cn (L.Q.); 2Aimei Vacin BioPharm (Zhejiang) Co., Ltd., Ningbo 315000, China

**Keywords:** mRNA, lncRNA, spleen, MPV-ACYW135, immune

## Abstract

The Group ACYW135 meningococcal polysaccharide vaccine (MPV-ACYW135) is a classical common vaccine used to prevent *Neisseria meningitidis* serogroups A, C, Y, and W135, but studies on the vaccine at the transcriptional level are still limited. In the present study, mRNAs and lncRNAs related to immunity were screened from the spleens of mice inoculated with MPV-ACYW135 and compared with the control group to identify differentially expressed mRNAs and lncRNAs in the immune response. The result revealed 34375 lncRNAs and 41321 mRNAs, including 405 differentially expressed (DE) lncRNAs and 52 DE mRNAs between the MPV group and the control group. Results of GO and KEGG enrichment analysis turned out that the main pathways related to the immunity of target genes of those DE mRNAs and DE lncRNAs were largely associated with positive regulation of T cell activation, CD8-positive immunoglobulin production in mucosal tissue, alpha-beta T cell proliferation, negative regulation of CD4-positive, and negative regulation of interleukin-17 production, suggesting that the antigens of MPV-ACYW135 capsular polysaccharide might activate T cell related immune reaction in the vaccine inoculation. In addition, it was noted that Bach2 (BTB and CNC homolog 2), the target gene of lncRNA MSTRG.17645, was involved in the regulation of immune response in MPV-ACYW135 vaccination. This study provided a preliminary catalog of both mRNAs and lncRNAs associated with the proliferation and differentiation of body immune cells, which was worthy of further research to enhance the understanding of the biological immune process regulated by MPV-ACYW135.

## 1. Introduction

Meningococcal polysaccharide vaccine (MPV) is a classical common vaccine used to prevent epidemic cerebrospinal meningitis, which is an acute bacterial respiratory infectious disease caused by *Neisseria meningitidis* (*N. meningitidis*) [1]. *N. meningitidis* is a kind of Gram-negative dicoccus, mostly arranged in pairs, that is parasitic in human nasopharyngeal mucosa, and humans are also the only host of the bacteria found so far [2]. *N. meningitidis* usually attaches to epithelial cells in the nasopharynx via pilus, and when human immunity is low, meningococcus will pass through the epithelium into the bloodstream via a series of outer membrane proteins and receptor molecules on the cell [3]. In the blood, meningococcus can cause a series of inflammatory responses that eventually cross the blood-brain barrier through a mechanism and finally cause meningitis [4]. In the cerebrospinal fluid of patients, most *N. meningitidis* bacteria are located in the neutrophils, with typical morphology [3]. The pathogenicity of meningococcus is related to its lipopolysaccharide, pilus, capsular polysaccharide, regulatory ferritin, outer membrane protein, IgA protease, and other virulence factors, among which the capsular polysaccharide (CPS) on the surface of the bacterial body has the effect of anti-phagocytosis and makes the bacteria invasive, which is the main pathogenic factor [2,5]. Meningitis can be treated with chemicals, such as beta-lactam antibiotics, and resistance is rarely reported [2]. However, meningitis develops rapidly, and even if timely treatment is given, death often occurs within 2 days of onset or serious sequelae are caused. In addition, the breakdown of the bacteria causes more endotoxins to be released into the bloodstream, with serious consequences [6,7]. Therefore, early preventive treatment is the ideal way to prevent meningitis. Thus, vaccination is the most cost-effective option.

According to the different structure and antigenicity of the capsular polysaccharide of *N. meningitidis*, which was divided into 13 serogroups, including A, B, C, D, X, Y, Z, 29E, W135, L, H, I, and K [2,5]. A total of 95% of severe meningitis and septicemia cases are caused by groups A, B, C, W135, and Y [8]. Meningococcal bacteria that invade the bloodstream produce large amounts of lipopolysaccharide, the main cause of fulminant infections [2,5]. In this study, MPV-ACYW135, which can effectively prevent the occurrence, transmission, and prevalence of meningococcal meningitis in groups A, C, Y, and W135, is selected to immunize mice. MPV-ACYW135 is made from group A, group C, group Y, and group W135 meningococcal culture media, and four kinds of serum meningococcal polysaccharide antigen are extracted and purified, respectively. After mixing, an appropriate stabilizer is added and frozen; the vaccine could stimulate the human body to produce immunity to the pathogen [9]. Over the past several decades, most studies indicated that the bacterial CPS was a non-T cell-dependent antigen; it was for this reason that it could not lead to the production of immune memory, and subsequently, it could only produce antibodies of the IgM type [2,5,10]. Although there have been many studies on the immune mechanisms of capsular polysaccharide antigens and immunological research on MPV in the past few decades, studies on the vaccine at the transcriptional level are still limited, especially in the study of long non-coding RNAs (lncRNAs).

LncRNA is one type of non-coding RNA without protein-encoding ability and longer than 200 bp in length, coming from regions of the transcriptome [11]. Previous research had confirmed that a number of lncRNAs could influence the expression of target genes by simply recruiting epigenetic complexes or impacting the transcription process [11,12,13]. More specifically, lncRNAs could segregate microRNAs or destabilize messenger RNA by recruiting several transcription factors to the target genomic DNA [13,14]. Moreover, lncRNAs might also serve as signals, guides, decoys, and scaffolds in cell biological processes based on their typical molecular mechanisms [13]. In addition, a number of previous studies have also indicated that lncRNAs could also act as crucial regulatory factors in numerous biological processes, including immunity [12,15], whereas how most lncRNAs affect the immune process remains unknown.

In this study, high-throughput transcriptome sequencing was performed to identify lncRNAs and mRNAs associated with immunity in mouse spleens between the MPV-ACYW135 group and the control group. The aim of this project was to investigate the potential role of immune-related lncRNAs in MPV-ACYW135 vaccination and further provide new insights into the molecular mechanisms of polysaccharide vaccines.

## 2. Materials and Methods

### 2.1. Animal Treatment and Sample Collection

MPV-ACYW135 was provided by Ningbo Aimei Vacin BioPharm (Zhejiang) Co., Ltd., Ningbo, China. All the laboratory mice were kept in the Laboratory Animal Center at Zhejiang University. A total of 24 (4-week-old) male healthy ICR mice were randomly selected and equally divided into two groups, the MPV group and the control group, with twelve experimental mice in each group. Each cage houses three mice, and all experimental mice could drink and eat freely. The feeding and management of the laboratory mice in the current study were approved by the Experimental AniTablemal Ethics Committee of Zhejiang University.

In addition, the procedure of MPV vaccination was as follows: all the 24 laboratory mice were accommodated in a breeding cage for three days, and then the MPV group was inoculated with MPV-ACYW135 according to their body weight on the third day, and the mice of the control group were inoculated with sterile, pyrogen-free PBS. Two weeks after vaccination, all the experimental mice were euthanized, and then spleen tissues and blood were collected.

The spleen was divided into several pieces and frozen in liquid nitrogen immediately to isolate RNA. In RNA-seq and RT-qPCR, three mice from each group were randomly selected, and each sequencing sample contained only one mouse. Serum was isolated from blood by centrifugation, and then the IgG, IgM, and IgA concentrations in serum were measured immediately by ELISA.

### 2.2. RNA Isolation, Library Preparation and Sequencing

The total RNA used to sequence and run RT-qPCR was isolated and purified from mouse spleens by the TRIzol method (see Appendix A). The purity and amount of the extracted RNA were quantified by NanoDrop2000. The integrity of extracted RNA was tested by the 2100 Bioanalyzer of Agilent with an RIN number > 7.0. The rRNA Removal Kit of Ribo-Zero™ was applied to remove the rRNA from the extracted total RNA. The remaining RNA was finally fragmented into numerous small fragments at a relative high temperature in the presence of divalent cations. Afterwards, cleaved RNA fragments were ultimately reversed into cDNA with a length of 300 ± 50 bp. At last, an Illumina Hiseq 4000 platform was adopted to carry out the paired-end sequencing at Lianchuan Biotechnology Co., Ltd., Hangzhou, China. The comprehensive workflow figure for RNA sequencing is shown in Figure 1.

### 2.3. Quality Control and Mapping

The transcriptome was sequenced by Illumina’s paired-end RNA-seq approach, which finally generated millons of 2 × 150 bp paired-end reads. Low-quality reads were filtered and subsequently discarded by Cutadapt (Version: cutadapt-1.9). Then, FastQC was used to confirm the quality of the sequences, including the GC content, Q20, and Q30 of the clean data. The raw sequence data had been submitted to GEO datasets with the accession number GSE224584. Both Bowtie2 and Hisat2 were applied to map all clean reads to the genome of *Mus musculus*, assembly GRCm39 NCBI [16,17]. After that, the mapped reads were finally assembled using StringTie. All transcripts obtained from the spleens of mice were reconstructed into a new comprehensive transcriptome by a Perl script. The expression levels of all the transcripts were finally estimated by StringTie [18] and Ballgown [14] according to the final reconstructed transcriptome.

### 2.4. Identification of lncRNA

First of all, transcripts < 200 bp or that overlapped with known mRNAs were filtered out. The Coding-Non-Coding Index (CNCI) and Coding Potential Calculator (CPC) were adopted to forecast the coding potential of transcripts [19,20]. At last, transcripts with a CNCI score less than 0 and a CPC score less than −1 were filtered out, and the remaining were regarded as lncRNAs.

### 2.5. DE mRNAs and DE lncRNAs Identification

The expression levels of the above-identified mRNA and lncRNA were analyzed by FPKM using StringTie. *p* < 0.05 and log2 (fold change) > 1 or log2 (fold change) < −1 were considered DE mRNAs and DE lncRNAs by DESeq2 software [21]. The significance threshold was derived from the adjusted *p*-values based on FDR.

### 2.6. Target Gene Prediction of lncRNAs and Functional Analysis

In this study, cis-target genes of lncRNAs were subsequently predicted to further explore their biological functions. The Python script was used to select the coding genes (100 kb upstream and downstream). Whereafter, the functional analysis of the identified target genes of lncRNAs in this study was performed by BLAST2GO [22].

### 2.7. Results of GO Enrichment and KEGG Enrichment Analysis

GO and KEGG enrichment analyses were subsequently carried out to investigate the biological processes of both mRNAs and lncRNAs, which were able to help further understand the biological function of the DE mRNA and DE lncRNAs in mice treated with MPV.

### 2.8. RNA Sequencing Result Validation by RT-qPCR

Six mRNAs (Igf2, Tsga13, Fbxo44, Zfp979, Myh4, and Myh1) and six lncRNAs (MSTRG.11204.1, MSTRG.13306.2, MSTRG.11959.2, MSTRG.2836.3, MSTRG.10054.2, and MSTRG.23771.1) representing the differential expression levels of RNA-seq were randomly selected from the spleens of 12 mice for RT-qPCR. The RT-qPCR was run by the SYBR Premix Ex Taq kit on an ABI Step One Plus instrument. The primers are listed in Appendix A. Moreover, the relative mRNA expression levels were finally normalized by the *β-actin* of mice, and the results were calculated by a 2^−ΔΔCt^ assay with 3 independent biological replicates [23]. In addition, the RT-qPCR measurements were all repeated in triplicate.

### 2.9. Statistical Analyses

Live weight, antibody levels in serum, and RT-qPCR results were compared using one-way ANOVA in SPSS 20.0 and finally correlated by Bonferroni. The data was expressed as “Mean ± SE”. *p* < 0.05 was regarded as statistically significant, and *p* < 0.01 was highly significant. The Shapiro–Wilk test was used to test the normal distribution of the data in SPSS 20.0.

## 3. Results

### 3.1. Phenotypic Data Analysis

A total of four phenotypic traits were measured and analyzed, including the live weight of mice during the whole experiment time and the concentration of IgG, IgM, and IgA in each mouse’s serum (n = 12). The live weight result showed that one day after the vaccine (or PBS) was administered, the mice in both groups experienced obvious weight loss, followed by slow weight gain, and there was no significant (*p* > 0.05) body weight change between the control group and MPV group (Figure 2). In addition, the MPV group had significantly higher (*p* < 0.01) humoral immune antibody levels concentrated in serum, including IgG, IgM, and IgA.

### 3.2. Sequencing Data Summary

The sequencing data obtained 84.85 GB of raw data from six libraries of mouse spleens. In more detail, the MPV group generated 96,472,780, 93,140,486 and 97,597,100 raw reads from each sample, whereas the control group generated 92,320,946, 91,885,218 and 94,280,934 raw reads from each sample. The above raw reads underwent extensive filtering to produce clean reads, which were subsequently mapped to the *Mus musculus* genome (GRCm39 version), with the final mapping ratio ranging from 94.32% to 94.78%. Appendix A displays a thorough summary of the sequencing data.

### 3.3. Identification of lncRNAs and mRNAs in Mouse Spleens

The analysis result in Appendix A shows that 34,375 putative lncRNAs were obtained in this study, including 28,108 known lncRNAs and 6727 novel lncRNAs. The genomic locations of the novel lncRNAs in this study were as follows: 3049 were intronic lncRNAs (48.65%), 1273 were intergenic lncRNAs (20.31%), 942 were bidirectional lncRNAs (15.03%), 574 were sense lncRNAs (9.16%), and 429 were antisense lncRNAs (6.85%). Figure 3A shows that the average length of the lncRNA transcripts identified in this study was shorter than the length of the mRNA transcripts. Moreover, 79.58% of lncRNAs contain four or fewer exons, while 79.90% of mRNAs contain three or more exons. Furthermore, Figure 3C,D showed that mRNAs found in the present study revealed longer open reading frames than lncRNAs in mouse spleen tissues.

### 3.4. Determination of DE mRNAs and DE lncRNAs

The FPKM levels were used to distinguish the DE mRNAs and DE lncRNAs in mouse spleens. Between the MPV group and the control group, a total of 52 DE mRNAs (Appendix A) and 405 DE lncRNAs (Appendix A) were discovered. 203 lncRNAs and 9 mRNAs were found to be significantly upregulated in the MPV group (*p* < 0.05), while 202 lncRNAs and 43 mRNAs were found to be significantly downregulated (*p* < 0.05). Figure 4A,B displays the volcano plots of the verified DE mRNAs and DE lncRNAs.

### 3.5. DE mRNA Functional Enrichment

The primary biological roles of the discovered DE mRNAs in this study were examined using GO analysis. 52 DE mRNAs were enriched for 596 GO terms with functional annotation (Appendix A). The result revealed 283 significant (*p* < 0.05) enriched GO terms, which were mainly enriched in extracellular region, ossification, collagen fibril organization, extracellular region, and biomineral tissue development (Figure 5A,B). In addition, KEGG pathway analysis revealed only eight significantly (*p* < 0.05) enriched pathways, such as Focal adhesion, ECM-receptor interaction, human papillomavirus infection, protein digestion and absorption, and the PI3K-Akt signaling pathway (Figure 5C). Detailed KEGG pathway information is listed in Appendix A.

### 3.6. Cis-Regulatory Functions of DE lncRNAs in Mouse Spleen Tissues

The cis-regulated target genes of the identified DE lncRNAs were predicted to further investigate their biological functions in mouse spleens. Appendix A shows 57 potential lncRNAs-mRNA pairs that were significantly correlated with 100kb as the cutoff. According to the outcome of DE lncRNA-gene pair prediction in cis-regulation by the values of the Pearson Correlation Coefficient, the leading five and final four lncRNA-mRNA pairs are listed in Table 1 below. The leading five lncRNA-mRNA pairings reveal the same regulatory orientations, whereas the final four pairs reveal the opposite.

Moreover, GO analysis found 4611 GO terms based on the cis-regulated target genes (Appendix A), including 77 significant terms (*p* < 0.05). The target genes of the identified DE lncRNA in this study were shown to be associated with negative regulation of ruffle assembly, modification of synaptic structure, response to superoxide, and DNA replication proofreading. The molecular functions were mainly focused on oxidative phosphorylation uncoupler activity and acetylgalactosaminyl-O-glycosyl-glycoprotein beta-1,6-N-acetylglucosaminyltransferase activity. In addition, the main cellular components were related to the protein phosphatase 4 complex and the mitochondrial oxoglutarate dehydrogenase complex (Figure 6A,B). KEGG pathway analysis indicated that the target genes of the discovered DE lncRNAs were primarily enriched for homologous recombination, SNARE interactions in vesicular transport, and phenylalanine, tyrosine, and tryptophan biosynthesis (Figure 6C, Appendix A).

### 3.7. Analysis of DE lncRNAs and DE mRNAs by Co-Enriched GO Terms

Five significant enriched GO terms (*p* < 0.05) were obtained from enrichment of both DE lncRNA and DE mRNA target genes in order to further explore the crucial pathways (Table 2). The significant co-enriched GO terms involved in immunoglobulin production in mucosal tissue are negative regulation of CD8-positive alpha-beta T cell proliferation, positive regulation of T cell activation, negative regulation of interleukin-17 production, and negative regulation of CD4-positive alpha-beta T cell proliferation, all of which belong to biological processes.

### 3.8. DE lncRNAs and DE mRNAs Verification Using RT-qPCR

Six DE lncRNAs and six DE mRNAs altogether were chosen, respectively, to confirm the RNA sequencing results by the RT-qPCR method. As shown in Figure 7, the results of RT-qPCR were consistent with the findings of RNA sequencing data, indicating that the RNA sequencing data was credible in our study, including both the transcript identification and abundance estimation. In addition, the MPV treatment group showed significantly higher (*p* < 0.01) relative mRNA expression levels of the mouse *Bach2* gene in the spleen than those of the control group (Figure 7C).

## 4. Discussion

Group ACYW135 meningococcal polysaccharide vaccine (MPV-ACYW135) is one of the most important vaccines used to prevent epidemic cerebrospinal meningitis, which is caused by *N. meningitidis* [1,2,5]. Meningitis develops rapidly and can cause serious sequelae [2,7]. Therefore, early preventive treatment is the ideal direction to prevent meningitis, and vaccination is the most effective response to such virulent diseases [1,2]. Positive immunization can significantly lessen the harm caused by avoidable illnesses while also increasing the efficiency with which medical resources are used under constrained circumstances [24]. So far, the meningococcal vaccine mainly contains polysaccharide vaccines based on the capsular polysaccharide antigen of *N. meningitidis* and polysaccharide-protein conjugate vaccines [1]. MPV-ACYW135 is a kind of polysaccharide vaccine that is made and purified by four kinds (groups A, C, Y, and W135) of serum meningococcal polysaccharide antigen of *N. meningitidis* [2,5,8]. The capsular polysaccharide of bacteria is a non-T cell-dependent antigen; it cannot cause helper T cell activation, nor can it stimulate Ig class conversion or immunological memory in B cells [10]. In this study, the live weight result showed that one day after the vaccine (or PBS) was administered, the mice in both groups experienced obvious weight loss, followed by slow weight gain. This might be caused by the injection, or it could be that the mice were stressed during the procedure. Nonetheless, there was no significant (*p* > 0.05) body weight change between the control group and MPV group during the experiment, suggesting that the vaccination of MPV-ACYW135 was not harmful to the mice’s health, at least in terms of weight gain. In terms of immunoglobulin, we found that the MPV group had significantly higher IgG, IgM, and IgA humoral immune antibody levels in serum. After the capsular polysaccharide antigens of *N. meningitidis* groups A, C, Y, and W135 enter the body for the first time, plasma cells will be produced, and antibodies will be synthesized and secreted after a certain incubation period. The first was IgM, but this antibody is short-lived and disappears quickly, lasting from days to weeks in the blood, and then IgG is produced next. During the immunity of MPV-ACYW135, IgM can bind to complement and is mainly distributed in serum. It is a highly effective anti-biological antibody due to its high binding valence. Its bactericidal, bacterolytic, phagocytic, and agglutinating effects are 500–1000 times higher than those of IgG. In the MPV-ACYW135 immunization, although the amount of IgM is not very large, as the first and most powerful antibody, IgM plays an important role in the early defense of the body [25]. IgG is the main component of human serum immunoglobulin, accounting for 70–75% of the total immunoglobulin, and it is the most durable and important antibody in the primary immune response. The result of IgG concentration in serum after MPV-ACYW135 inoculation in this study was consistent with previous research; of all the antibodies, it has the highest absolute concentration. Most antibacterial antibodies belong to IgG, which plays a major role in anti-infection and can promote the phagocytosis of monocyte macrophages. This is the main reason why, after capsular polysaccharide antigens of MPV-ACYW135 enter the body, IgG antibody levels rise rapidly [26]. IgA can mediate the conditioning of phagocytic antibody-dependent cytotoxic effects mediated by cells and is the main component of the body’s mucosal defense system, covering the surface of the mucosa. It is an important mucosal barrier, acting as the first line of defense to prevent pathogens from invading the body and inhibiting the adhesion of microorganisms, slowing down the propagation of viruses [27]. Although MPV-ACYW135 has been used for several decades, the specific molecular mechanisms related to immunity remain limited, especially at the transcriptional level, such as lncRNA and mRNA.

The spleen is a key peripheral immune organ and the center of humoral and cellular immunity. As blood flows through the spleen, it can recognize pathogens and antigens, which stimulate various receptors in the spleen cells to activate the innate immune response [28]. As a consequence, the spleen was selected as the target organ for RNA-seq to analyze the transcription profiles of MPV-ACYW135-inoculated mice. In this study, we identified 34,375 lncRNAs and 41,321 mRNAs, including 405 DE lncRNAs and 52 DE mRNAs, between the MPV group and the control group. CPC and CNCI were used to predict the coding of potential transcripts. CPC is based on sequence alignment, which can facilitate protein-coding transcript selection but impairs the performance of noncoding transcripts. In addition, CPC is a time-consuming process. CNCI is designed to distinguish between long noncoding and coding transcripts without the annotation of sequences. A number of lncRNAs are poorly annotated, which allows for more accurate discrimination of these sequences. CNCI shows acceptable results on vertebrates (except fish); however, for invertebrates and plants, the results are not very satisfying. CNCI appears to be superior to CPC, with the sequences becoming longer [29].

Numerous studies implicated lncRNAs as important regulators in a large number of biological processes, including immunization [15,30]. The present study revealed comprehensive lncRNA and mRNA profiles of mouse spleens inoculated with MPV-ACYW135 for the first time. The findings showed that the obtained lncRNAs had shorter lengths of transcript and fewer exons than mRNAs, which agreed with previous reports among different species, suggesting that the identification of lncRNAs in this study was reliable. Moreover, the average expression levels of identified lncRNAs were significantly higher (*p* < 0.05) than those of mRNAs, indicating that the lncRNAs might be crucial for the immunization of mouse spleens.

LncRNAs may regulate the neighboring mRNA’s expression levels by the mode of coactivation or repression; in addition, lncRNA expression was highly associated with the expression of neighboring mRNAs [31]. In consequence, we speculated that the identified lncRNAs found in this study could significantly affect immune function by regulating the expression of their respective target mRNAs. In our study, DE cis-target genes were used to predict putative biological roles linked to immune modulation in mouse spleens. These genes were found within 100 kb upstream or downstream of the identified 405 DE lncRNAs. The predicted result displayed that DE lncRNA MSTRG.17645 may influence the DE coding gene *Bach2*. In addition, *Bach2* was significantly up-regulated (*p*< 0.05) in the MPV group. The full name of Bach2 is BTB (broad-complex, tramtrak, and bric a brac) and CNC (cap ‘n’ collar) homolog 2; it is a transcriptional suppressor with a wide range of functions in the regulation of immune cell differentiation [32]. Moreover, Bach2 and its paralog, Bach1, could regulate the immune responses in B cells, T cells, and innate immune cells [33]. Moreover, they work together to regulate the early cell development stages of B cells and T cells [34]. Their developmental roles in immune cells include inhibiting genes that contribute to the development of myeloid cells in progenitor cells, thereby promoting progenitor cell differentiation into B cells, T cells, and erythroid cells [35,36,37]. For B cells, Bach2 is required at every stage of B cell development to delay antibody-producing plasma cell differentiation and class switch recombination [38]. For T cells, Bach2 inhibits the differentiation of CD4(+) T cells into Th2 cells, suppresses the production of Th2 cytokines, and promotes regulatory T (Treg) cell generation and function to balance immune activities [39]. The effects of Bach2 on the development of B cells and T cells and their responses to antigens are of great importance not only for deeply understanding the fundamental mechanisms of humoral immunity and cellular immunity but also for understanding pathological processes such as those in autoimmune diseases and for further developing more effective vaccines that have lasting effects against pathogens. In addition, the RT-qPCR method was used in this study to verify the DE mRNA and DE lncRNA, and the results showed that the RNA sequencing data was credible in our study. The relative mRNA expression levels were calculated by 2^−ΔΔCt^ assay and normalized by *β-actin*. 2^−ΔΔCt^ assay assumes 100% efficiency of qPCR [40]. Although only one single reference gene was considered ideal in the past [23], a number of recent studies have reported that, as a reference gene, *β-actin* is stably expressed in mouse spleens and could be a single reference gene [41].

The majority of bacterial CPS antigens were previously thought to be T cell-independent and could not lead to helper T cell activation and immune memory in B cells, stimulate Ig class switching, or activate active processes and antigen presentation with MHC (major histocompatibility complex) molecules [2,5,10]. In consequence, polysaccharide vaccines such as MPV-ACYW135 are believed to activate humoral immunity only, and the main role of humoral immunity is assumed to be B cells [42]. Interestingly, the analysis results of GO terms and KEGG pathways show that there were several items associated with T cells, and the surprising results were similar to some research findings in the last few years [43,44]. Research on the biological function and immune-modulating properties of bacterial CPS has revealed that these compounds could activate T cells, promote the growth of T cells, dendritic cells, and other immune cells, and strengthen the biological function of immune cells by fostering their maturation and differentiation [2,5,10]. The capsular polysaccharide A (PSA) of *B. fragilis* could directly work as a T-cell antigen to regulate T-cell activity [44]. PSA could interact with the α-β T cell receptor to stimulate CD4+T cells to produce IL-10 [45]. Moreover, it was reported that PSA could also combine with Toll-like receptor 2 of T cells to directly affect FoxP3+ Tregs, promoting IL-10 production while inhibiting helper T cell 17 (Th17) response and interleukin-17 (IL-17) production [28,46]. Most of the previous research reported that all polysaccharides were T-cell-independent antigens. They could bind to surface immunoglobulin molecules and then crosslink and activate B cells, leading to a number of subsequent changes that allow B cells to differentiate into plasma cells and subsequently produce antibodies [10]. Nevertheless, in combination with the above evidence and the findings of our study, we speculated that T cells in spleens might be involved in the immune regulation of MPV-ACYW135 vaccination. The specific molecular mechanisms of how T cells in spleens participate in the immune regulation induced by CPS of MPV-ACYW135 remain to be further investigated.

In conclusion, the present study was the first to comprehensively describe the lncRNA and mRNA profiles of mouse spleens inoculated with MPV-ACYW135, and several DE lncRNAs and DE mRNAs were identified to be associated with immune regulation in mouse spleens after inoculating MPV-ACYW135. In addition, both DE lncRNAs and DE mRNAs obtained in our study could provide a new perspective for further research of the molecular immune mechanisms of MPV-ACYW135 at the transcriptional level. Last but not least, the lncRNA MSTRG.17645 may have a significant regulatory effect on its potential target gene, *Bach2*, after vaccination by MPV-ACYW135. It is necessary to do additional molecular and cellular research in order to validate the sequencing data.

## Figures and Tables

**Figure 1 vaccines-11-01295-f001:**
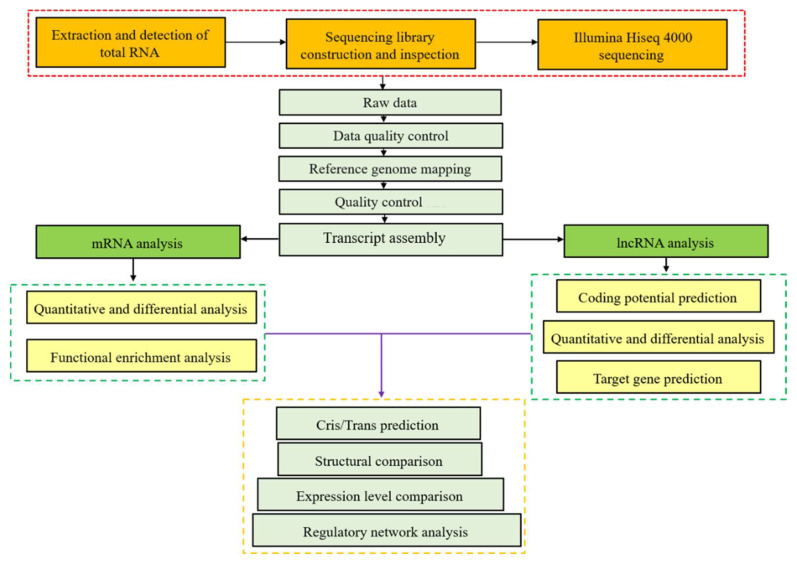
Comprehensive workflow figure for RNA sequencing.

**Figure 2 vaccines-11-01295-f002:**
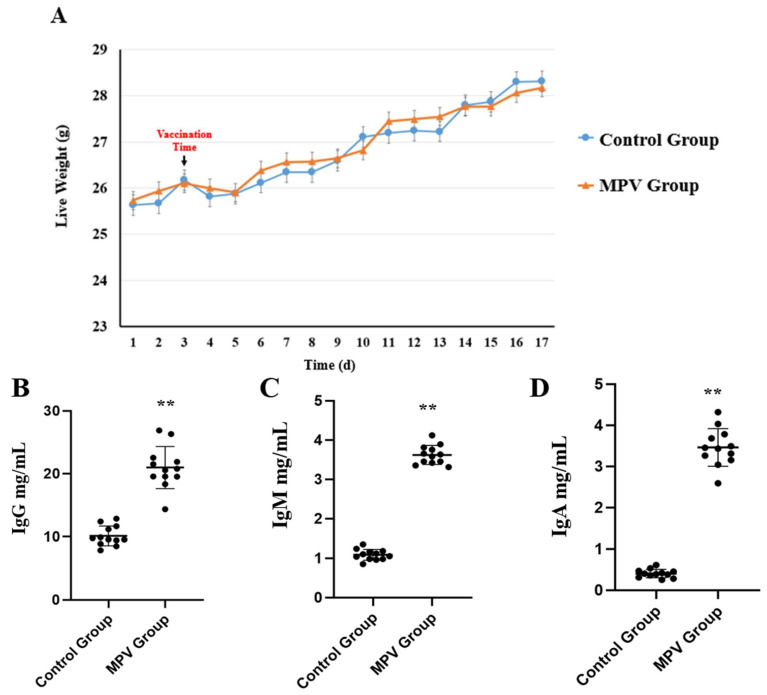
Live weights and serum antibody levels were compared between the MPV and control groups (n = 12). (**A**) Live weight comparison. (**B**) IgG level in serum. (**C**) IgM level in serum. (**D**) IgA level in serum. Significant differences are indicated by ** (*p* < 0.01).

**Figure 3 vaccines-11-01295-f003:**
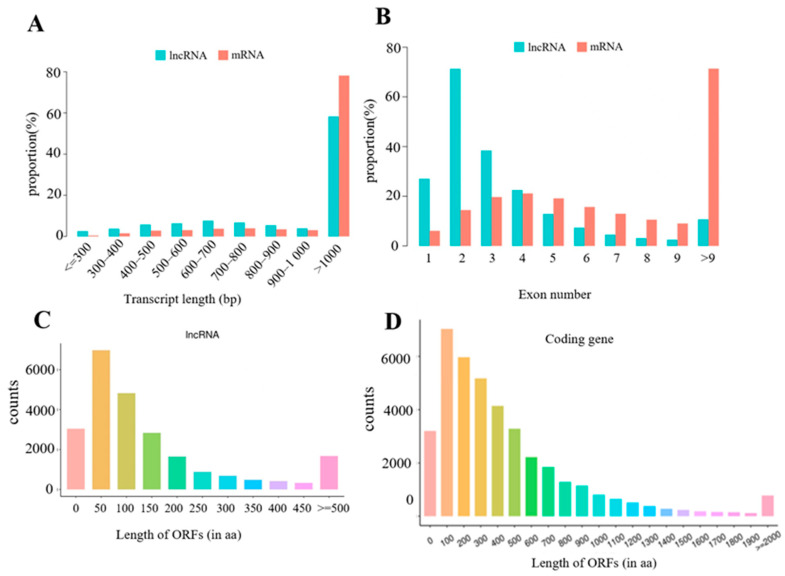
Genomic features of the obtained lncRNAs. (**A**) Transcript length distributions of obtained lncRNAs and mRNAs. (**B**) Exon number distributions of obtained lncRNAs and mRNAs. (**C**) Length of ORF distributions of obtained lncRNAs. (**D**) Length of ORF distributions of obtained mRNAs.

**Figure 4 vaccines-11-01295-f004:**
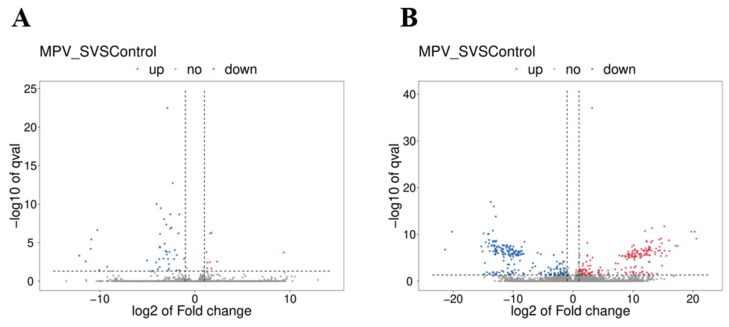
Volcano plots of DE mRNAs and DE lncRNAs between MPV and the control group. (**A**) Volcano plot of DE mRNAs. (**B**) Volcano plot of DE lncRNAs. Red points are for highly up-regulated mRNAs or lncRNAs, whereas blue points are considerably down-regulated.

**Figure 5 vaccines-11-01295-f005:**
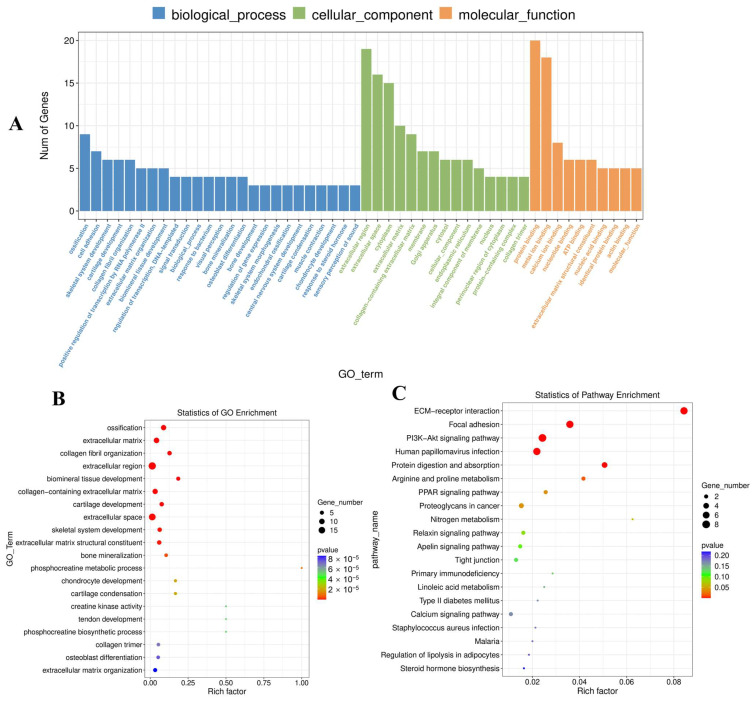
GO and KEGG analysis results of DE mRNAs. (**A**) GO enrichment result of DE mRNAs by histogram. (**B**) GO enrichment result of DE mRNAs by scatter plot. (**C**) KEGG enrichment result of DE mRNAs by scatter plot.

**Figure 6 vaccines-11-01295-f006:**
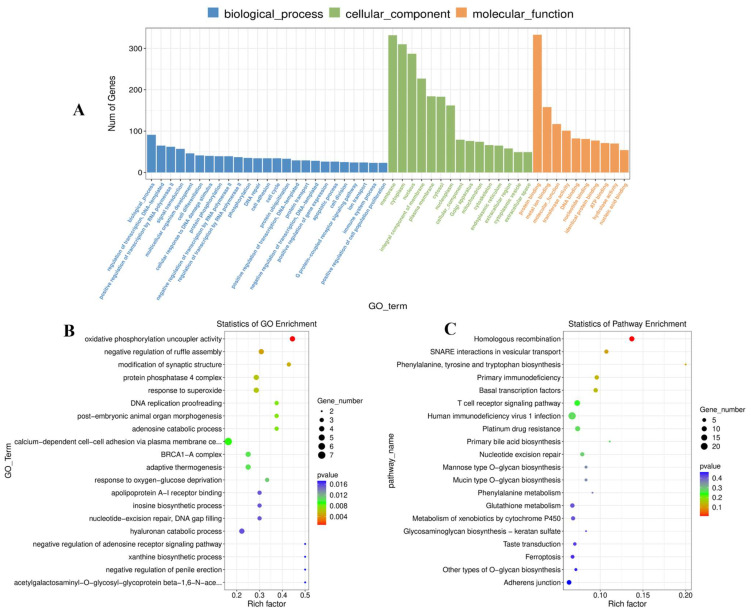
Results of DE lncRNA analysis using GO and KEGG (**A**) The histogram of GO enrichment results for DE lncRNAs. (**B**) A scatter plot showing the GO enrichment results for DE lncRNAs. (**C**) A scatter plot showing the KEGG enrichment results for DE lncRNAs.

**Figure 7 vaccines-11-01295-f007:**
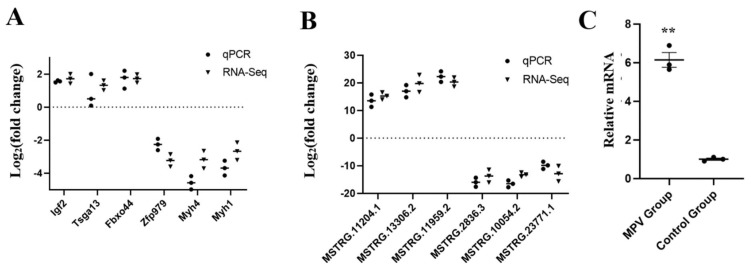
The verification result by RT-qPCR. (**A**) The validation of RT-qPCR with six mRNAs (n = 3). (**B**) The validation of RT-qPCR with six lncRNAs (n = 3). (**C**) Comparison of the relative mRNA expression levels of *Bach2* in the spleens of sequenced mice between the MPV group and the control group (n = 3). Significant differences are indicated by ** (*p* < 0.01).

**Table 1 vaccines-11-01295-t001:** DE lncRNA-mRNA pairs between the MPV group and the control group (cislocation: 100 kb).

Gene Name	lncRNA Transcript Name	Pearson Correlation Coefficient
Zfp384	ENSMUST00000203287	1
Fus	ENSMUST00000128851	1
Acy1	ENSMUST00000187798	0.93
Selenop	MSTRG.9430.8	0.90
Igf2	ENSMUST00000136359	0.90
Tmem106a	MSTRG.5353.1	−0.20
Slc30a5	MSTRG.8116.2	−0.20
Slc30a5	MSTRG.8116.4	−0.20
Ube2d3	ENSMUST00000181619	−0.19

**Table 2 vaccines-11-01295-t002:** Co-enriched GO terms of DE lncRNA and DE mRNA.

GO Term	GO Function	*p*-Value
immunoglobulin production in mucosal tissue	biological process	0.00
negative regulation of CD8-positive, alpha-beta T cell proliferation	biological process	0.04
positive regulation of T cell activation	biological process	0.02
negative regulation of interleukin-17 production	biological process	0.04
negative regulation of CD4-positive, alpha-beta T cell proliferation	biological process	0.02

## Data Availability

The data presented in this study are available in Appendix A here.

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
