# Peer review of "Comparative Transcriptome Profiling of mRNA and lncRNA of Mouse Spleens Inoculated with the Group ACYW135 Meningococcal Polysaccharide Vaccine"

_vaccines, 2023, doi:10.3390/vaccines11081295_

Round 1

Reviewer 1 Report

In this study, the authors sought to profile the transcriptional profiles of mice after injection with meningococcal vaccine MPV-ACYW135.  The speens of vaccinated or PBS control mice were analyzed two weeks after injection by RNA-seq with validation by qPCR, and a number of differentially expressed coding and noncoding genes were observed.  Many of the DE genes were across several GO term pathways, however, a subset related to immune responses overlapped between coding and noncoding RNAs.  Overall, the study is well designed and an intriguing and relevant investigation.  However, there are a few concerns regarding the methods and classification described below:   There is concern as lncRNA correlating with Fus (ENSMUST00000128851) in Table 1 is listed on Ensembl as a Fus transcript (Fus-206) but does not code for a protein due to a retained intron.  This brings up concern that perhaps other lncRNAs as called in this study are actually non-coding splice variants derived from known protein-coding genes rather than lncRNA genes, the authors should check that their hits are indeed from separate gene units.   There should be a clear text description about how many mice are in each sequencing sample.  Were all 12 mice in each group used in RNA-Seq with 4 mice pooled in each sample, or just three mice from each group?   Figure 2: Weight is misspelled.  It would be better to put the units in parentheses to make it seem less like a formula, ie. (g) or (d)   Figure 3:  the text of the axes in A and B are difficult to read.  D, for mRNAs, is mislabelled at lncRNA.  C and D could also be combined into one figure using the color scheme of A and B.   Figure 4 may be unnecessary as it does not provide information about the conclusions of the manuscript regarding transcription after the vaccine.   Line 244: is this 89?  The space between the numbers is confusing   Figures 6 and 7 are also difficult to read, these may be better off as full-page figures.  These figures are arguably distracting, as many of these terms are unrelated, and the primary conclusions are drawn from the GO terms that overlap in mRNA and lncRNA DE genes.  Therefore the authors may consider just leaving them as supplementary information.   Table 1 would look better if (bp) was shifted to the second line entirely instead of with the lone parentheses.  The distance column is unnecessary if 100kb cutoff is stated in the text, would it not be better to put the actual distance between the genes?   How are the p values in Table 2 calculated?  Are the DE genes from each term in mRNA and lncRNA reused as a combined input?   How many mice for each group are used in the qPCR for Figure 8?  Is there a reason Bach2 has its own figure rather than being combined into Figure 8?     Bach2 and potential regulation by lncRNA MSTRG.17645 take up a considerable amount of the discussion, but these are missing in the relevant analysis of Table 1. Italicize et al. throughout the document

Line 63: place a comma after W135 to maintain consistency with the Oxford comma

Line 83: place a space after Mallory Line 85: microRNAs does not need to be hyphenated Line 89:  immunity would be better than immune Line 114: Elisa should be capitalized to ELISA Throughout:  "mice spleen" should be made to "mice spleens" or "murine spleen" Figure 1: the and in "Quality Control and" is unnecessary.  "Transcripts assembling" might read better as "Transcript assembly" Lines 140, 200: Italicize Mus musculus Line 142: Change to "A perl script" Lines 194, 195: Date should be data Line 205: change the comma after follows to a colon Line 221:  remove the period after Figure 4 Lines 250 and 276: place a space after (B) Line 281: Co- does not need to be capitalized Line 340: change er to et Line 352: It might be better to change the sentence to read "analyze the transcription profiles of MPV-ACYW135-inoculated mice." Line 422: capitalize the y

Author Response

There is concern as lncRNA correlating with Fus (ENSMUST00000128851) in Table 1 is listed on Ensembl as a Fus transcript (Fus-206) but does not code for a protein due to a retained intron.  This brings up concern that perhaps other lncRNAs as called in this study are actually non-coding splice variants derived from known protein-coding genes rather than lncRNA genes, the authors should check that their hits are indeed from separate gene units.  

Thank you for the valuable suggestion. We examined it and found that the transcript ENSMUST00000128851 is not an independent locus, but indeed an intron-retained RNA transcribed by the gene Fus. Actually, we have checked all the lncRNA transcript, the result was shown in the excel files (Column M: trans_type).

There should be a clear text description about how many mice are in each sequencing sample. Were all 12 mice in each group used in RNA-Seq with 4 mice pooled in each sample, or just three mice from each group?  

In RNA-seq, three mice from each group were randomly selected, and each sequencing sample contains only one mouse. The description has been added in line 121-122.

Figure 2: Weight is misspelled.  It would be better to put the units in parentheses to make it seem less like a formula, ie. (g) or (d)  

It has been revised in Figure 2.

Figure 3:  the text of the axes in A and B are difficult to read.  D, for mRNAs, is mislabelled at lncRNA.  C and D could also be combined into one figure using the color scheme of A and B.  

Figure 3 has been modified, the text of the axes in A and B has been enlarged, and D has been revised. C and D are difficult to combined into one figure for the horizontal coordinate difference is a little big.

Figure 4 may be unnecessary as it does not provide information about the conclusions of the manuscript regarding transcription after the vaccine.

Figure 4 has been removed according to your comment.

Line 244: is this 89?  The space between the numbers is confusing  

Sorry for the confusion. KEGG pathway analysis revealed only 8 significantly (P<0.05) enriched pathways. It has been revised in line 257.

Figures 6 and 7 are also difficult to read, these may be better off as full-page figures.  These figures are arguably distracting, as many of these terms are unrelated, and the primary conclusions are drawn from the GO terms that overlap in mRNA and lncRNA DE genes.  Therefore, the authors may consider just leaving them as supplementary information.  

The two figures have been re-edited for readability, and both figures were uploaded in the supplementary information.

Table 1 would look better if (bp) was shifted to the second line entirely instead of with the lone parentheses.  The distance column is unnecessary if 100kb cutoff is stated in the text, would it not be better to put the actual distance between the genes?

Table 1 has been modified according to your suggestions.

How are the p values in Table 2 calculated? 

The p values in Table 2 were calculated by the following formula.

Here N is the number of all genes with GO annotation; n is the number of DEGs in N; M is the number of all genes that are annotated to the certain GO terms; m is the number of DEGs in M. N stands for Total background gene (TB gene number); n stands for Total significant gene (TS gene number); M stands for Background gene (B gene number); m stands for Significant gene (S gene number). GO terms meeting this condition with p < 0.05 were defined as significantly enriched GO terms in DEGs. This analysis was able to recognize the main biological functions that DEGs exercise.

How many mice for each group are used in the qPCR for Figure 8? 

Three mice for each group are used in the qPCR for Figure 8, and it has been added in line 307.

Is there a reason Bach2 has its own figure rather than being combined into Figure 8?    

It has been combined into one figure according to your suggestion.

Italicize et al. throughout the document

It has been revised throughout the document.

Line 63: place a comma after W135 to maintain consistency with the Oxford comma

It has been revised in line 66.

Line 83: place a space after Mallory

It has been revised in line 88.

Line 85: microRNAs does not need to be hyphenated

It has been revised in line 91.

Line 89:  immunity would be better than immune

It has been revised in line 95.

Line 114: Elisa should be capitalized to ELISA

It has been revised in line 122.

Throughout: "mice spleen" should be made to "mice spleens" or "murine spleen"

"mice spleen" have been made to "mice spleens" throughout the manuscript.

Figure 1: the and in "Quality Control and" is unnecessary. "Transcripts assembling" might read better as "Transcript assembly"

It has been revised in Figure 1.

Lines 140, 200: Italicize Mus musculus

It has been revised in line 150 and 212.

Line 142: Change to "A perl script"

It has been revised in line 152.

Lines 194, 195: Date should be data

It has been revised in line 206 and 207.

Line 205: change the comma after follows to a colon

It has been revised in line 217.

Line 221:  remove the period after Figure 4

It has been revised in line 235.

Lines 250 and 276: place a space after (B)

It has been revised in line 262 and 287.

Line 281: Co- does not need to be capitalized

It has been revised in line 292.

Line 340: change er to et

It has been revised..

Line 352: It might be better to change the sentence to read "analyze the transcription profiles of MPV-ACYW135-inoculated mice."

It has been revised in line 384.

Line 422: capitalize the y.

It has been revised in line 463.

Reviewer 2 Report

The Manuscript (vaccine-2428648) entitled “Comparative transcriptome profiling of mRNA and lncRNA of mice spleens inoculated with the group” The authors adopted an approach to identify the mechanism of mRNAs and lncRNAs on the immune response in response to MPV-ACYW135 vaccine. The description, methodology, results, and discussion described appropriate and scientifically sound. However, many grammatical and typographical errors need to be corrected as some of those mentioned below. The manuscript should revise by a native English speaker. I recommend this MS to be published in vaccines after minor revision and thoroughly proofreading.

Minor:

Line 89: including immune? or immunity or immune responses or immune mechanisms. 

Line 100: re-write, A total of 24 4-weeks-old, e.g. A total of 24 (4-weeks-old) male ICR mice.

Line 103: free water and feed; free means?

Line 114: Elisa should be capitalized ELISA.

Line 129: correct generatd “generated”

Line 244: KEGG pathway analysis revealed only 8 9?   89 or 8.9 or 8 and 9?

Line 327-329: check the sentence if it is grammatically correct.

Line 372: The predicted result instead of the Prediction result.

Line 343: It is an important mucosal barrier, is the first line of defense.?

Line 344-345: rewrite the sentence.

Line 407: differentiation of them?

Increase the readability of Figures 3, 6 and 7.

In the last line of the conclusion: correct the typographic errors like researched and sequencing date. And what kind of molecular and cellular research you suggested for further studies?

A native English speaker should revise the manuscript.

Author Response

Minor:

Line 89: including immune? or immunity or immune responses or immune mechanisms. 

It has been changed into immunity in line 93.

Line 100: re-write, A total of 24 4-weeks-old, e.g. A total of 24 (4-weeks-old) male ICR mice.

It has been revised in line 103.

Line 103: free water and feed; free means?

It has been changed into “all experimental animals could drink and eat freely”.

Line 114: Elisa should be capitalized ELISA.

It has been revised in line 118.

Line 129: correct generatd “generated”

It has been revised in line 133.

Line 244: KEGG pathway analysis revealed only 8 9?   89 or 8.9 or 8 and 9?

Sorry for the confusion. KEGG pathway analysis revealed only 8 significantly (P<0.05) enriched pathways. It has been revised in line 250.

Line 327-329: check the sentence if it is grammatically correct.

This sentence has been checked and revised in line348-351.

Line 372: The predicted result instead of the Prediction result.

It has been revised in line 376.

Line 343: It is an important mucosal barrier, is the first line of defense.?

This sentence has been revised in line 369.

Line 344-345: rewrite the sentence.

The sentence has been rewrited in line371-373.

Line 407: differentiation of them?

Sorry, I'm a little confused by this comment. The capsular polysaccharide could promote maturation and differentiation of T cells and dendritic cell. I changed “them” to “these immune cells”?

Increase the readability of Figures 3, 6 and 7.

Figures 3, 6 and 7 have been modified.

In the last line of the conclusion: correct the typographic errors like researched and sequencing date. And what kind of molecular and cellular research you suggested for further studies?

They have been revised in line 457-458. Vectors with knockdown and overexpression of the identified lncRNAs will be constructed to transfect spleen cells or immune cells to further verify the signaling pathway and regulatory mechanism.

Reviewer 3 Report

I revised the manuscript. I made some corrections to the article. The article should be revised for English language by an native Engilish speaker and references according to journal rules.

The manuscript will contain concise information and add new ideas to the literature.

 The article should be revised for English language by an native Engilish speaker 

Author Response

The manuscript has been revised according to your comment, and the English language has been modified by a native English speaker.

Reviewer 4 Report

The work of Zhu N., titled "Comparative transcriptome profiling of mRNA and lncRNA of mice spleens inoculated with group ACYW135 meningococcal polysaccharide vaccine". It is a very interesting work, but requires adding some information.

L110.-Please explain, the reason for inoculating the control group with PBS and not with the vehicle of the MPV-ACYW135 vaccine?

L111.-Taking into consideration that the authors,  only studied spleen tissues and blood.  Explain, if the lncRNAs studied correspond to ubiquitously express or tissue-specific lncRNA,  and did not consider other tissues that could have a greater proportion of lncRNA, how does humans happen?

L148, L151.-Indicate all abbreviations the first time they appear, such as: Coding Potential Calculator (CPC),  Coding-Non-Coding Index (CNCI), Differentially expressed (DE).

L249.- Improve Figure 6

L252.- The authors should show also, the results of the trans-, and Ce regulation networks that could also predict which DE mRNAs might be regulated by DE lncRNAs.

L274.- Improve Figure 7 B and C

L331-344.- This discussion of the immune response is very general, it should be described in terms of the immune response to MPV-ACYW135

L301.- In discussion indicate the limitations of the use of CPC and CNCI in the lncRNA Identification. (Review Han S. DOI: 10.1155/2016/8496165)  and,  also indicate the general limitations of the work.

Author Response

The work of Zhu N., titled "Comparative transcriptome profiling of mRNA and lncRNA of mice spleens inoculated with group ACYW135 meningococcal polysaccharide vaccine". It is a very interesting work, but requires adding some information.

L110.-Please explain, the reason for inoculating the control group with PBS and not with the vehicle of the MPV-ACYW135 vaccine?

The MPV-ACYW135 vaccine is made from the polysaccharide antigens extracted and purified from the culture medium of N. meningitidis group A, C, Y and W135, and then mixed and lyophilized. The vaccine is white loose body, which is colorless and clear liquid after being redissolved with the included diluent. The vaccine diluent is sterile, pyrogen free PBS. The MPV-ACYW135 vaccine belongs to polysaccharide vaccine, rather than polysaccharide conjugate vaccine. Therefore, the control group was inoculatd with sterile, pyrogen free PBS.

L111.-Taking into consideration that the authors, only studied spleen tissues and blood.  Explain, if the lncRNAs studied correspond to ubiquitously express or tissue-specific lncRNA, and did not consider other tissues that could have a greater proportion of lncRNA, how does humans happen?

This is a very instructive comment, and that's a limitation of our study. Many studies have reported that lncRNA was tissue-specific and spatiotemporal specific. LncRNA expression levels were different in different tissues, and the expression levels of lncRNA in the same tissue or organ might also change at different growth stages. It is hard to judge that the lncRNAs we found in this study correspond to ubiquitously express or tissue-specific lncRNA according to the present results.Therefore, in the following functional verification experiment, we will give full consideration to the comments of your comment, considering other tissues that could have a greater proportion of lncRNA.

L148, L151.-Indicate all abbreviations the first time they appear, such as: Coding Potential Calculator (CPC), Coding-Non-Coding Index (CNCI), Differentially expressed (DE).

Thank you for your reminder. All abbreviations the first time they appear have been indicated in the manuscript.

L249.- Improve Figure 6

Figure 6 has been improved in the manuscript.

L252.- The authors should show also, the results of the trans-, and Ce regulation networks that could also predict which DE mRNAs might be regulated by DE lncRNAs.

We did not show the results of the trans-, because previous studies indicated that lncRNA's trans target gene prediction was only suitable for large sample sizes, and if the sample size was too small (such as less than 6), the analysis will not be reliable. Ce regulation networks is a very valuable analysis, but there are only two types of RNA (mRNA and lncRNA) in this study, which is a little difficult for Ce regulation networks.

L274.- Improve Figure 7 B and C

Figure 7 B and C have been improved in the manuscript.

L331-344.- This discussion of the immune response is very general, it should be described in terms of the immune response to MPV-ACYW135

Thank you for the valuable comment. This is very helpful for us to improve the quality of our article. New discussion has been described in the manuscript in line 353-353.

L301.- In discussion indicate the limitations of the use of CPC and CNCI in the lncRNA Identification. (Review Han S. DOI: 10.1155/2016/8496165) and, also indicate the general limitations of the work.

It has been added in line 387-395 in the manuscript.

Round 2

Reviewer 4 Report

The authors have done a good job addressing my comments, I have no further suggestions. I think the paper is acceptable for publication in Vaccines.

Author Response

Article has been revised.